# An Individual’s Connection to Nature Can Affect Perceived Restorativeness of Natural Environments. Some Observations about Biophilia

**DOI:** 10.3390/bs8030034

**Published:** 2018-03-05

**Authors:** Rita Berto, Giuseppe Barbiero, Pietro Barbiero, Giulio Senes

**Affiliations:** 1Department of Human Sciences, University of Verona, via San Francesco 22, 37129 Verona, Italy; 2Department of Human and Social Sciences, Laboratory of Affective Ecology, University of Valle d’Aosta, Strada Cappuccini 2/a, 11100 Aosta, Italy; g.barbiero@univda.it; 3IRIS, Interdisciplinary Research Institute on Sustainability, University of Torino, Via Accademia Albertina, 13, 10123 Torino, Italy; 4Department of Mathematical Sciences (DISMA), Politecnico di Torino, Corso Duca degli Abruzzi 24, 10129 Torino, Italy; pietro.barbiero@studenti.polito.it; 5Department of Agricultural & Environmental Sciences, University of Milano, via Celoria 2, 20133 Milano, Italy; giulio.senes@unimi.it

**Keywords:** perceived restorativeness, connection to Nature, biophilia, biophilic quality

## Abstract

This study investigates the relationship between the level to which a person feels connected to Nature and that person’s ability to perceive the restorative value of a natural environment. We assume that perceived restorativeness may depend on an individual’s connection to Nature and this relationship may also vary with the biophilic quality of the environment, i.e., the functional and aesthetic value of the natural environment which presumably gave an evolutionary advantage to our species. To this end, the level of connection to Nature and the perceived restorativeness of the environment were assessed in individuals visiting three parks characterized by their high level of “naturalness” and high or low biophilic quality. The results show that the perceived level of restorativeness is associated with the sense of connection to Nature, as well as the biophilic quality of the environment: individuals with different degrees of connection to Nature seek settings with different degrees of restorativeness and biophilic quality. This means that perceived restorativeness can also depend on an individual’s “inclination” towards Nature.

## 1. Introduction

With this research we seek to investigate the relationship among the extent to which people feel connected to Nature, the person’s ability to perceive the restorative value of the natural environment, and the *biophilic quality* of that environment [1,2], which can be roughly summarized in the environment’s naturalness, functional, and aesthetic value. In this paper, “Nature” is written with a capital “N” to indicate the biosphere and the abiotic matrices (soil, air, water) where it flourishes, and to avoid confusion with “nature” as the intrinsic quality of a certain creature and/or phenomenon. This study is part of a broader project, the goal of which is to gain a better understanding of how people become aware of the positive benefits associated with exposure to natural environments in aiding restoration from mental fatigue, and eventually whether, and how, the restoration process can affect how people care for Nature (see [3,4]).

### 1.1. Perceived Restorativeness from the Evolutionary Perspective

The construct “perceived restorativeness” (PR) is a by-product of the Attention Restoration Theory (ART) [5]. According to ART, what makes an environment “restorative” is the limited need for directed attention and Nature’s capacity to entice involuntary attention; the former is the kind of attention that requires effort, whereas the latter does not. As W. James stated: “The best attention is effortless” [6]. Unfortunately, most situations encountered during the day call for voluntary directed attention and the price of all this effort is mental fatigue. Mental fatigue implies that the mechanism responsible for inhibiting distractions, and which voluntary attention depends upon, is saturated. Indeed, distractions (such as concurrent thoughts and competitive stimuli) are abundant in daily life and they must be inhibited for voluntary attention to function efficiently. The activation of mechanisms able to restore directed attention capacity is, therefore, fundamental; one such way is via exposure to natural environments [5]. Direct contact with Nature mainly activates bottom-up involuntary attention, and since people are not required to focus on specific “less interesting” stimuli in natural environments, energy does not need to be directed towards suppressing such “distracting” stimuli (for a more in-depth discussion see [7,8,9]). In ART, this type of involuntary effortless attention has been referred to as “fascination” and the capacity to generate fascination in the observer is the most important characteristic of a restorative environment [5]. Natural settings engage fascination, thus, directed attention can rest and be restored from mental/attentional fatigue, and this underlies the restorative quality of Nature. In this regard, an individual’s perceived restorativeness can represent an important evolutionary adaptation that corresponds to the assessment of the opportunity the environment may offer to recover direct attention from mental fatigue [3].

### 1.2. Why Do Certain Individuals Seem Less Able to Perceive the Restorative Properties of Nature?

However, only some people consider, and can appreciate, exposure to natural environments as an effective and cost-free way of recovering from one’s daily hassles. Indeed, the average person is generally unaware of the psychological benefits that can be gained from immersing oneself in Nature and this “lack” of awareness may affect the perception of the restorativeness associated with exposure to Nature. Indeed, some works have shown that many people do not seem to perceive the restorative qualities of Nature and, thus, do not generate a preference for such environments [10,11]; whereas, other people’s experiences of Nature actually enhance their preference for natural environments [12] and, therefore, their willingness to spend time in this type of environment. The reasons why certain individuals may be less able to perceive the satisfying, restorative properties of Nature are unclear, but it may partly be caused by a lack of positive “immersive” experiences in the past [13]. 

The positive effects of Nature may depend on the degree of immersion [14]; in other words, the effects of Nature exposure may be more robust when individuals are more fully immersed in these environments and more fully “present” in their context compared with when they are exposed to “reproduced” Nature and distracted by thoughts and/or external stimuli unrelated to the natural environments. When immersed, individuals more fully attend to the characteristics of their surroundings, they recognize and contact more aspects of the natural environment [15]. Supporting research into virtual contexts found that immersion in real world environments led to greater memory formation of these environments [16] and correlated with higher enjoyment associated with engaging with them [17]. Moreover, immersion may trigger a “fuller” experience and, thus, a more robust reaction to natural (or non-natural) stimuli. Ample evidence has substantiated the positive effects of interacting with Nature upon health and well-being; evidence has come from studies focused on the effects of outdoor activities, the therapeutic use of Nature, access to views of Nature (real or reproduced), and the use of plants in indoor environments (for a review see [18]); the findings of which all substantiate the biophilia hypothesis [19,20,21,22]. 

### 1.3. Biophilia Implies Affective Connection to Nature

According to E.O. Wilson, biophilia is innate and cemented in our evolutionary history during which we developed a set of genetically-determined learning rules [23] (p. 31). Biophilia steers our relationship with Nature, including environmental preference. Although wild animals and human beings are guided by instinct in choosing their own habitat, there are some differences. For wild animals the habitat search is oriented by a genetically determined and very little modifiable instinct (see e.g., [24]). In our species, habitat search is guided by instinct [25], but human instinct consists mainly of learning rules [23]. For this reason, the relationship between humans and natural environments has also been shaped, to some extent, by cultural and individual factors, such as empirical experiences and acculturation [26,27]. 

In our species, environmental preference is related to the aesthetic quality of the natural environment; biophilia implies affection towards animals, plants, and all other living organisms, as well as an innate preference for natural environments because of our evolutionary past ([28] (pp. 176–178), [29]). Although not all aesthetic characteristics can be traced back to evolutionary adaptations, among them it is not difficult to recognize those which originally were survival rules [25]. Humans have been evolving over the past 200,000 years in natural environments; they have grown and reorganized in response to the natural environments and, indeed, become fascinated by them [30]. Having an apparent “connection with Nature” (CN) reflects the degree to which individuals believe themselves to be a part of the natural world [31], i.e., the belief that we belong to it as much as it belongs to us. Studies into the connection or association people experience with Nature have intensified over the last few years [32,33]. These studies propose that the ability to “connect” with Nature is a positive personality characteristic that improves cognitive capacity, emotional well-being, positive mood, and happiness. This is due because humans are genetically programmed to function effectively in natural environments and because there is evidence for genetically-determined biases that affect environmental preference [34]. As said above, biophilia is cemented in our evolutionary history as well; it is affected by the ability to focus on natural stimuli effortlessly [35] (p. 134), i.e., the capacity to be fascinated by Nature [5], and by asymmetric empathy, i.e., to engage emotionally with the various lifeforms and to be affected by their perceived condition ([26,28] (pp. 49–53)). Research demonstrates that the more time people spend in natural environments engendering greater asymmetric empathy towards Nature, the more they feel an increasing sense of connectedness to it [36,37,38,39]. People who have greater experiences of the natural environment may express greater affective connection to it than those with less experience. 

### 1.4. Towards a Definition of “Biophilic Quality” of the Environment

The literature suggests that the perceived restorative quality of an environment depends on the specific characteristics of the setting; on the contrary, connection to Nature has been proposed to be a characteristic of the subject and not determined by the setting. In this research study we propose that the level to which a person feels connected to Nature might determine that person’s “ability” to perceive the restorative value of a natural environment. Accordingly, we implicitly assume that restorativeness may depend on an individual’s connection to Nature and this relationship may vary with the *biophilic quality* of the environment, its biophilic appeal. The biophilic quality is the set of characteristics which makes the environment “objectively” restorative based on humans’ evolutionary adaptation rules, bypassing an individual’s assessment of restorativeness [1]. Biophilic quality and perceived restorativeness are expected, to some extent, to match. However, this relationship may vary with an individual’s sense of connection that appears to be an emergent property of people interacting with natural environments that are pleasing both aesthetically and functionally [18], i.e., endowed with the biophilic quality [2]. This attachment facilitates the vision of an interaction between “form” and “function”, because together they stimulate progressively stronger emotions towards the environment; as one becomes increasingly attached to the biophilic quality of an environment one can be said to be engaged with it [40]. It is not necessarily the case that we are aware of the reasons for our attachment (genetic predisposition) [25,28,29], but we undoubtedly become effortlessly and unconsciously connected to environments that support our informational needs (making sense, exploring solutions for adaptation), and steered towards psychological benefits (e.g., stress recovery and attentional restoration). Despite our individual differences, we share a similar mental model that recognizes which environmental contexts are supportive and adaptive [40,41,42]. From the evolutionary point of view, biophilia is the shared mental model [22] and humans’ predisposition to recognize the biophilic quality of a certain habitat reflects the adaptations designed by natural selection aimed to help us to choose the place where to live [27,43,44]. 

### 1.5. What Is the Role of Familiarity in Perceived Restorativeness?

Differences in perceived restorativeness mainly result from differences in the landscapes evaluated and from the environmental preferences of individuals (see [17]); research has found a positive correlation between environmental preference and perceived restorativeness and a lack of any correlation between familiarity and perceived restorativeness (see, e.g., [12,45]). Furthermore, research using the Perceived Restorativeness Scale found that the higher restorative values of natural versus urban or artificial settings did not differ with gender or age [46,47,48]. Recently, people’s connection to Nature was identified as an antecedent of positive perceptual experiences of natural settings that predict the perceived level of restorativeness of a landscape and the sense of safety, coherence and complexity it conveyed; in brief, individuals with a stronger connection to Nature were significantly more likely to assess a forest setting as having a stronger restorative potential than individuals with a weaker connection to Nature [49]; this result was independent of the individuals’ familiarity of the natural settings assessed. Although the literature reports no significant correlation between perceived restorativeness and familiarity of an environment (see [12,45,46]), we believe that familiarity is likely to play a role in the perceptual evaluation of natural landscapes because a person with a deeper connection to Nature may also be more familiar with natural environments; indeed, connection to Nature and/or Nature appreciation has been shown to result from knowledge of and frequent contact with natural environments [36,50]; additionally, familiarity can be predictive of environmental preference [51]. For this reason, the present study takes familiarity into account, as well as perceived restorativeness and connection to Nature. In addition to that, as people may well react differently when in an actual natural environment compared with when simply observing the image of an environment (see [14,52]), our subjects were tested during visits to real natural settings. 

## 2. The Aim of this Study

To the study’s aim, four different natural areas were chosen and classified based on the Recreational Opportunity Spectrum (ROS) [53]; this system was originally devised for classifying and managing recreation opportunities based on the following criteria: physical, social, and managerial setting. The ROS is a means of identifying and determining the diversity of recreation opportunities for a natural area and it is based on the idea that visitor services quality is best assured by providing an array of opportunities suited to the full range of expected visitors, where opportunities are carefully scheduled considering that not all visitors seek the same experience or activities when they visit a natural area. Maes et al. [54] tested the ROS classification thorough four thematic pilot studies to assess UE wilderness areas per ecosystem type and ecosystem services, where the ROS was combined with: (1) the degree of naturalness identified as a proxy for people’s preference for more natural areas; (2) protected areas as public recreation areas and as providers of recreation services and facilities; (3) water attractiveness. Lately the ROS was implemented by Paracchini et al. [55] who merged the Recreational Potential Indicator (RPI) for the EU areas in terms of “remoteness” and “accessibility”; the potential for recreation has been classified in three classes of high-medium-low provision (Figure 1). This new classification was framed by a broad set of key policies, and structured around a conceptual framework that linked human wellbeing to the environment [55]. 

The ROS can be considered an indicator of the natural value of a setting [54,55,56]; since at the time of this study no instrument was available to assess the biophilic quality (BQ) of the environment (now it is the *Biophilic Quality Index* [1]), we assume the ROS classes may roughly correspond to the biophilic quality of an environment, where a low biophilic value corresponds to classes 1–4, and a high biophilic value to classes 5–9. 

The method used in this study to investigate how connection to Nature and perceived restorativeness are related and eventually vary with the biophilic quality of the environment may appear somehow counterintuitive; in fact, only settings with a high degree of naturalness and high biophilic quality were considered. The reasons for this are two-fold. First, according to [40], emotional attachment to an environment is stronger in settings that are both aesthetically and functionally pleasing. Second, perceived restorativeness varies according to an environment’s degree of naturalness [12]. However, the present study is not interested in these variations; on the contrary, we want to look for differences in the level of connectedness to Nature in people visiting settings with a high biophilic quality. The literature shows that connection to Nature is a stable trait in adults [36,57], therefore, as no variations in connectedness are expected to occur between settings characterized by different degrees of naturalness and/or from moment to moment, and we can safely assume that the level of connection to Nature measured in each subject during the park visits in our study faithfully reflects that subject’s level of connection to Nature. 

Furthermore, by studying only environments high in naturalness, we can investigate the variations in the levels of connection to Nature and/or perceived restorativeness. Evidence in the literature suggests restorativeness to be a characteristic of the place, whereas connection to Nature belongs to the subject; but how can we be sure of this? We hypothesize that a high sense of connection to Nature helps an individual to perceive the restorative benefits of Nature, and we implicitly assume that restorativeness may also depend on the individual’s connection to Nature; at the same time, we hypothesize that a high sense of connection to Nature can be triggered by the specific characteristics of an environment, and we implicitly assume that certain environments make us feel more connected to Nature. 

## 3. Method

### 3.1. Settings

Three Italian protected natural areas and an urban park were chosen as settings for this study: Riserva della biosfera Valle del Ticino (a biosphere reserve), Parco nazionale della Val Grande (a national park), Parco naturale dell’Alpe Veglia e dell’Alpe Devero (a natural park). A fourth setting, Parco di Trenno (a peri-urban park in Milano, recently renamed “Parco Aldo Aniasi”), was used as the control condition (Figure 2). From here on, we will use the following abbreviations: Trenno (for Parco Aldo Aniasi), Ticino (Riserva della biosfera Valle del Ticino), ValGrande (for Parco nazionale della Val Grande), Devero (for Parco naturale dell’Alpe Veglia e dell’Alpe Devero). 

The four parks were classified according to the ROS classes and assigned the corresponding biophilic quality value (Table 1). 

### 3.2. Participants

A total of 524 subjects older than 18 years of age were approached on-site; of those, 435 (83%) accepted to participate in the research study: 239 males and 196 females, aged 18–85 years (M = 44.17 years, SD = 16.96). The participants were chosen using a convenience sampling procedure.

Only Italian visitors of the four above-listed settings were recruited into the study (Table 2) and interviewed at the end of their visits. 

### 3.3. Constructs and Measures

Subjects were administered a questionnaire that required them to assess their experience of the natural setting using the following three scales: (i) the Perceived Restorativeness Scale-11 (PRS-11) to assess the setting perceived restorativeness with two additional items included to assess familiarity and preference; (ii) a list of physical and aesthetic attributes to assess the setting; and (iii) the Connectedness to Nature Scale to assess subject’s sense of connection to Nature. Participants were also asked to provide their age, gender, and residential location (for more details on the questionnaire see [58]). The questionnaire and the scales were in Italian.

*Perceived restorativeness (PR).* The PRS-11 [59] based on the original version by [60] measures an individual’s perception of four restorative factors (alpha = 0.89) [60], (alpha = 0.87) [2]: *being-away* (a setting that allows physical and/or psychological distance from demands on directed attention); *fascination* (the type of attention stimulated by interesting objects, namely a setting that provokes curiosity in the individual and fascination about things and assumed to be effortless and without capacity limitations); *coherence* (a setting where activities and items are ordered and organized); and *scope* (a setting that is large enough such that it does not restrict movement, thereby offering a sort of “world of its own”). Items are rated on a 0 to 10-point scale, where 0 = not at all, 6 = rather much, and 10 = completely. 

*Familiarity and preference (FAM and PREF)*. Familiarity (one item: “this place is familiar to me”, from the original PRS) [60] and preference (one item: “I like this place”, from PRS-11) [59] were also measured. Both items are rated on a 0 to 10-point scale, where 0 = not at all, 6 = rather much, and 10 = completely. 

*A list of physical and aesthetic attributes.* A list of attributes, already shown to be reliable in previous studies, was used to assess subjects’ perception of the following sensorial and symbolic aesthetic attributes [61]: Vegetation, visual diversity/richness, harmony/congruence, openness/spaciousness, luminosity, representativeness, cleanliness, maintenance/upkeep, place for leisure activities, meeting place, and novel place (alpha = 0.83) [2], plus accessibility, safety, tranquility, crowdedness, and artificiality (alpha = 0.75) [58]. All attributes are rated on a 1 to 5-point scale, where 0 = not at all, and 5 = a lot.

*Connection to Nature (CN).* The CNS [37] measures the extent to which people feel a part of the natural world, i.e., a sense of oneness with the natural world, a sense of kinship with animals and plants and a sense of equality between self and Nature. The scale is made up of 14 items and judgments are made on a 1 to 4-point scale, where 1 = never and 4 = always (alpha = 0.84) [37], (alpha = 0.82) [58]. 

### 3.4. Procedure

The same procedure was used for each setting. The on-site administration was conducted during a six-week period from early August to mid-September 2015, on weekdays and weekends and under the same sunny weather condition. Times of day and day of the week were counterbalanced in order not to under or over represent certain types of visitors.

In the first place, participants were given a general overview of the study, they were asked to answer a questionnaire made up of general questions about the visit and the park characteristics and of a few scales aimed to assess how they experience the natural setting, for approximately 10–15 min. There was only one researcher administering the survey and he remained available during and after the completion of the questionnaire for any additional questions.

Participants’ inform consent was obtained and confidentiality was guaranteed. 

## 4. Results

For each setting, the mean PRS-11 scores were calculated (considering the summed overall score for all 11 items; from here on PRS), for preference (PREF) and familiarity (FAM). Similarly, the mean CNS scores were also calculated (Figure 3). These variables served both the traditional statistics and the neural pattern recognition analysis run on the data. 

### 4.1. Statistical Analyses

#### 4.1.1. Multivariate Analysis of Variance

A MANOVA (multivariate analysis of variance) was run to investigate the effect of each setting (fixed factor, 4 levels) on PRS, CNS, PREF, and FAM mean scores. A significant effect of setting emerged for all variables: PRS F(3, 431) = 29.88; CNS F(3, 431) = 4.38; PREF F(3, 431) = 10.22; FAM F(3, 431) = 20.87. All effects were significant at the *p* < 0.001 level. The same analysis was run on the physical and aesthetic attribute scores (Table 3); again, a significant effect of setting emerged for all variables, *p* < 0.05. Dependent variable scores differentiated significantly across settings, following roughly this decreasing order: Devero, ValGrande, Trenno/Ticino, reflecting the biophilic quality level of the four settings. 

#### 4.1.2. Pearson’s Correlation

To assess the direction and strength of the relationship between the variables addressed in this study, a Pearson’s correlation was run considering all subjects and all settings together. As reported in Table 4, all correlations were significant and the overall trends for the correlations between PRS × FAM and PRS × CNS were found to be highly similar (Table 4). 

A Pearson’s correlation was then run on the same pairs of variables but considering each setting separately (Table 4). The significant correlations revealed between PRS × PREF and between PREF × FAM for all the settings were not unexpected; on the other hand, this study is the first to show a significant correlation between PRS × FAM. As far as the correlation between PRS × CNS is concerned, it varied across settings: it was strongest in relation to Devero, followed by Trenno, and then ValGrande, whereas no correlation was present in relation to Ticino. The strength of the correlation between PRS × CNS for each setting appeared to follow the strength of the biophilic quality value assigned to that particular setting and the trend of the physical aesthetic attributes (Table 3). For Ticino and Trenno, no correlation between FAM × CNS was observed (Table 5). 

A significant role of FAM, the other independent variable besides connection to Nature, cannot be ruled out. Considering the correlation results, FAM may affect both perceived restorativeness and preference. When we look carefully at the sample characteristics (for more details see [58]), we notice that a large proportion of subjects visiting Devero (52.78%) and ValGrande (37.04%) drove more than 100 km to reach the setting, whereas the average distance travelled was less than 10 km with respect to Ticino (43.86%) and Trenno (89.52%). Despite the long distance involved to reach Devero and ValGrande, we found that those visiting these two destinations were highly familiar with the settings.

#### 4.1.3. Linear Regression Analysis

A linear regression with curve estimation was then performed to attempt to assess and model the relationship between PRS vs. CNS and PRS vs. FAM. Keeping PRS as the dependent variable, when CNS is the predictor: R^2^ = 0.121 (adjusted R^2^ = 0.119); and when FAM is the predictor: R^2^ = 0.115 (adjusted R^2^ = 0.113). Despite the low values, both linear regressions produced the same significant result: PRS increases in parallel with CNS and FAM. Now a stepwise multiple regression was run. Once again, PRS was taken as the dependent variable, but now two predictors were considered (CNS and FAM) to observe which one would be eliminated by the model. The model excluded FAM and generated plots that were highly distinct to those obtained for the linear regression. The first plot shows a linear relationship between PRS and CNS, whereas the second only reveals a cluster of high FAM–high PRS values, although no linear relationship was apparent.

#### 4.1.4. Kolmogorov-Smirnov Test

Figure 4 shows the differences among the four settings according to the data collected on the PRS. At a first glance, those settings with low BQ (Trenno and Ticino) seem to have a quite similar PRS distribution as well as the settings with high BQ (ValGrande and Devero); but if you look more carefully low BQ settings and high BQ setting distributions differ significantly showing the PRS distribution lowers on the left only for low BQ settings (see Figure 4). Basically, PRS histograms have different traits: settings with low biophilic quality (Figure 4, upper plots) tend to have a higher variance and a lower mean of PRS than settings with high BQ (bottom plots). The Kolmogorov-Smirnov test run on these data corroborated the existence of at least two different types of settings from the PRS point of view (*p* < 0.001): high vs. low. Taking Figure 4 together with Figure 5, they show that Trenno/Ticino and ValGrande/Devero come from two distributions.

### 4.2. Neural Pattern Recognition

At this point a Multi-Layer Perceptron (MLP) was used to test more precisely whether CN affects PR of a natural setting with a given BQ value. The MLP is a mathematical model, it is a simple but rather robust Artificial Neural Network (ANN) used for pattern recognition [62,63]. To perform a binary classification, data were transformed from discrete into binary. Basically, the MLP allows answering whether a nonlinear relationship exists between connection to Nature (CN) and perceived restorativeness (PR), estimating subject’s PR on the subject’s level of CN. More specifically, the MLP was trained using the discrete approximation for the PRS, where PR is “low” when the setting scores lower than 7, whereas it is “high” when the score is higher than 7 (Berto, 2005). 

The ANN was trained on 50% of the data collected from the assessment instruments, while 25% was used to check the training validity. At the end of the training process, the MLP learnt to estimate a high or low PR using the information concerning the subject’s CN level, i.e., the neural network knows the correlation between the variables. At this point the ANN was asked to make an educated guess on the remaining 25% of the data. The MLP correctly recognized the subject’s PR in 75% of the cases. By the way, if we consider only individuals with a high sense of connection to Nature (in our sample individuals with a low sense of connection to Nature were underrepresented) the correlation among the variables is recognize in 97% of the cases. 

The MLP has a different “granularity” than the statistical analysis, this means that its results are less refined, but more reliable. In this regard, given the reliable outcomes from the MLP, the previous statistical results were confirmed to be consistent. 

## 5. Discussion

The literature suggests that the perceived restorative quality of an environment depends on the specific characteristics of the setting (for this reason the PRS-11 does not include “compatibility” among the restorative factors; see [59]); on the contrary, connection to Nature has been proposed to be a characteristic of the subject and not determined by the setting. However, we believe that things are not so clear-cut. In this study, we observed that people with a high sense of connection to Nature seem to better perceive the restorative benefits of Nature. This can implicitly mean that perceived restorativeness also depends on an individual’s “inclination” toward Nature. However, if a high sense of connection to Nature can be triggered by an environment’s characteristics, then it also stands that certain environments will make us feel more connected to Nature. 

The significant effect of the setting emerging from the MANOVA upon PR, PREF, and FAM is easy to comprehend for these measures, but it becomes more complex regarding CN if we consider connectedness to be a characteristic of the subject. As already stated in the introduction, no data were collected regarding the participants’ perceived levels of connection to Nature prior to their excursions in the four settings; however, the literature argues this construct to be stable in adults over time. We can, therefore, make the following two remarks about this result: (1) if we assume that the level of connection to Nature measured within the setting reflects the exact level of connection to Nature of the individual, then individuals characterized by greater connection to Nature are more likely to visit settings with a higher restorative value and biophilic quality, i.e., more natural, functional, and aesthetically pleasing characteristics; and (2) if we consider the level of connection to Nature to be strictly linked to the setting’s characteristics (as in the case of perceived restorativeness and preference), then the extent to which individuals feel connected to Nature depends on the characteristics of the environment, i.e., on the biophilic quality. As such, an extremely beautiful and highly-restorative setting would be expected to make people feel more connected to Nature. However, these two explanations are apparently incongruent, as we will see in the next section. 

The results suggest that individuals with different degrees of connection to Nature seek settings with different degrees of restorativeness and biophilic quality. However, is this plausible? Due to the lack of baseline data for connection to Nature, we are unable to confirm this statement unequivocally; however, it is likely that an individual’s level of connection to Nature may affect their perception of the restorative characteristics of the setting. The significant differences regarding the perceived restorativeness assessed between the control setting (Trenno) and the other settings, all of which were characterized by a high level of naturalness, confirmed our expectations; whereas the differences among the three experimental settings belonging to the same environmental category were unexpected. Once again, is it the individual’s underlying connection to Nature that determines their perception of the restorative characteristics of a natural setting? If we refer to Figure 3 (disregarding the control condition), we notice that the trends for CN, PR, and PREF are indistinguishable between the different settings.

Moving on to the correlation results, and in particular to the relationship between PR and CN, the most interesting finding regards the control setting (Trenno: a peri-urban park). The visitors attending Trenno were also those with the lowest levels of connection to Nature (Figure 2). Did this occur by chance or is it plausible that people with low connection to Nature are more likely to visit settings with low biophilic quality? It would seem to make more sense to think that the physical characteristics of the setting induce different levels of connection to Nature; i.e., they may enhance the sense of connection to Nature in some individuals or, on the contrary, even suppress this sense of connection in others (see remark number 2 above). It is more difficult, on the other hand, to suggest an explanation for the absence of a correlation between PR and CN for the Ticino setting. Table 3 shows that the mean scores for the physical and aesthetic attributes were lowest for this setting, indicating this park to be the least “spectacular” over Devero and ValGrande. Two plausible explanations can be proposed for this result: (1) the subjects visiting the Ticino are characterized by a good (prior) level of connection to Nature, and accordingly may have expected more from Ticino from a biophilic and restorative point of view (Figure 3); and (2) Ticino lacks certain characteristics (Table 3) necessary to be fully appreciated and properly fit the individuals’ level of connection to Nature. Both explanations lie more in favor of remark number 2, i.e., that connection to Nature is strictly linked to an environment’s characteristics. 

A final observation is reserved for familiarity; although connection to Nature and familiarity with the natural environment sometimes overlap (as our results show), they are two different aspects of the same experience, i.e., direct and frequent exposure to Nature. The results show that we were right not to disregard the association of the FAM on PR scores; familiarity (FAM) and the capacity to perceive restorative factors (as assessed by the PRS) are not independent of one another, as the literature has suggested. Our strategy of interviewing in the field may have allowed this result to emerge (i.e., due to the immersion effect); it is likely that the effect of FAM assessed in the field merged into the effect of an individual’s connection to Nature and what was actually measured was the participants’ familiarity for natural environments, in general. In fact, the strength of the correlation between PR and FAM and between PR and CN came out to be the same (Table 4); moreover, linear regression showed identical trends for the effects of CN and FAM on PR. 

## 6. Some Observations about Biophilia

Here, connection to Nature is assumed to reflect human ontological evolution in relation to Nature, i.e., a subject’s personal history with Nature, therefore, meaningful experiences with Nature turn to a high sense of connection to Nature. On the other hand, the capacity to perceive restorativeness may reflect our phylogenetic history as a species. Over the course of the past 200,000 years of *H. sapiens*’ evolution, natural selection has strongly shaped our characteristics and humans have learnt that certain environments can aid recovery from attentional fatigue and psycho-physiological stress more than others (for a review see) [18]. The African savannah was the original environment offering this restorative opportunity and there is much discussion over the potential benefits to be gained from reproducing and encouraging people to engage with that kind of environment today (Savannah hypothesis) [64]. From this standpoint, since humans are genetically predisposed to respond positively to certain environments only, then perceived restorativeness can be considered a measure of human innate biophilia [22]. Berto, Pasini, and Barbiero [47] recently found that children’s perceived restorativeness increases during the course of a day experiencing a woodland environment, whereas their sense of connection to Nature does not. This is another reason we should consider perceived restorativeness as an “immediate” response (due to our genetic predisposition) [23] to an environment that we perceive to be restorative or not; whereas generating significant changes in connection to Nature requires a specific educational course that takes place over a longer period. Indeed, the human species is characterized by slow enculturation, where culture prevails over instinct [27,65]. 

Our results gain further meaning when we consider the biophilic quality (BQ; for more details see [1,2,66]): it is likely that a subject only perceives the restorative quality of an environment to be high when he/she engages with an environment characterized by biophilic quality that fits his/her level of connection to Nature. If this speculation is proven true, then the relationship between CN, PR, and BQ can be summarized as shown in the Figure below (Figure 6), where a subject’s perceived restorativeness (PR) is the product of two factors: the individual’s connection to Nature (CN) and the environment’s biophilic quality (BQ).

## 7. Concluding Remarks

The literature has argued that connection to Nature is a characteristic of the individual, being independent of the environment to which they are exposed, whereas the level of perceived restorativeness of a natural setting depends entirely on that environment’s characteristics. However, the results of the present study show a less clear-cut situation worthy of further investigation. This study was inspired by the fact that a deeper appreciation of and affiliation with natural environments is a potential antecedent of positive perceptual experiences in natural settings and that individuals with higher levels of connection to Nature view natural landscapes as more attractive and fascinating; in other words, the subjective sense of connection to Nature is associated with more positive perceptual evaluations of natural settings [49]. Our study not only confirmed the relationship between connection to Nature and aesthetic appreciation, but it also showed that the biophilic quality of the environment can be associated to the levels of connection to Nature and perceived restorativeness. Although the literature reports connection to Nature and perceived restorativeness to be independent phenomena, our results tell a different story, indicating that they are in fact intertwined and associated with the biophilic quality of the environment; a subject’s perceived restorativeness is only high when he/she is exposed to an environment characterized by the biophilic quality that exactly fit his/her own level of connection to Nature. 

This study has attempted to shed further light on the curious interactions occurring between people and restorative natural environments; it goes beyond the mere perceptual assessment of the restorative characteristics of a setting and considers, first, the role played by a subject’s connection to Nature; and, second, the fit between the biophilic quality of a setting and a subject’s connection to Nature on perceived restorativeness. Unfortunately, the absence of baseline measurements for connection to Nature may be a limit of this study; in fact, we could not verify whether subjects’ connection to Nature really remained stable (as supposed in adults) or instead increased or even decreased because of their visit. Another limitation, which could represent a threat to construct validity, was using one item to operationalize familiarity and preference, though they were taken from the PRS original version [60] and from the PRS-11 [59], respectively. However, the main limit of this study concerns the assessment of the setting biophilic quality, since at the time of the study this construct had no corresponding measurement instrument. However, the use of indirect measures, related to this construct, allowed us to speculate and draw important conclusions. We hope to solve this issue soon thanks to the Biophilic Quality Index, the instrument that allows rating objectively how an environment is biophilic [1]. Nevertheless, our results are encouraging in that they point towards a virtuous loop between perceived restorativeness, connection to Nature, and environment biophilic quality. 

## Figures and Tables

**Figure 1 behavsci-08-00034-f001:**
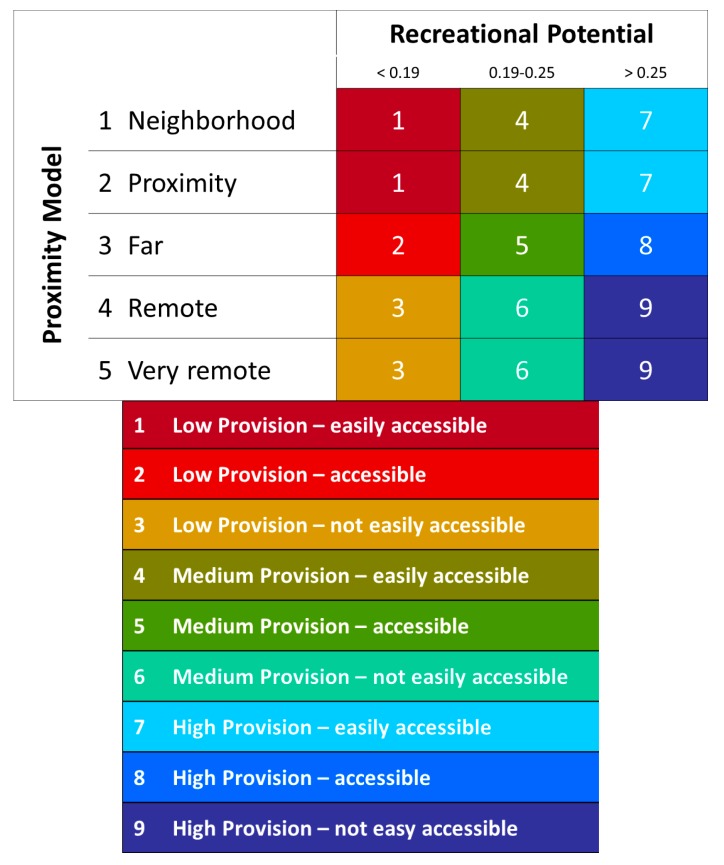
The Recreation Opportunity Spectrum (ROS) classes according to Paracchini et al. [55]. In this study, ROS served to classify the biophilic quality (BQ) of the settings: a ROS value from 1 to 4 indicates a setting with low BQ; a value from 5 to 9 indicates a setting with high BQ.

**Figure 2 behavsci-08-00034-f002:**
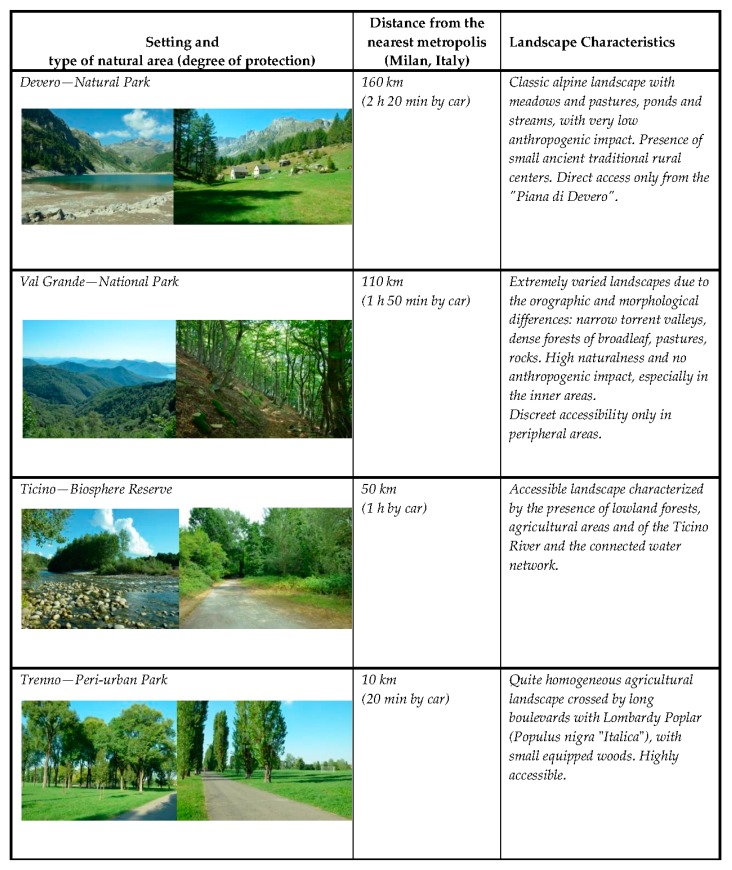
Description of the four settings.

**Figure 3 behavsci-08-00034-f003:**
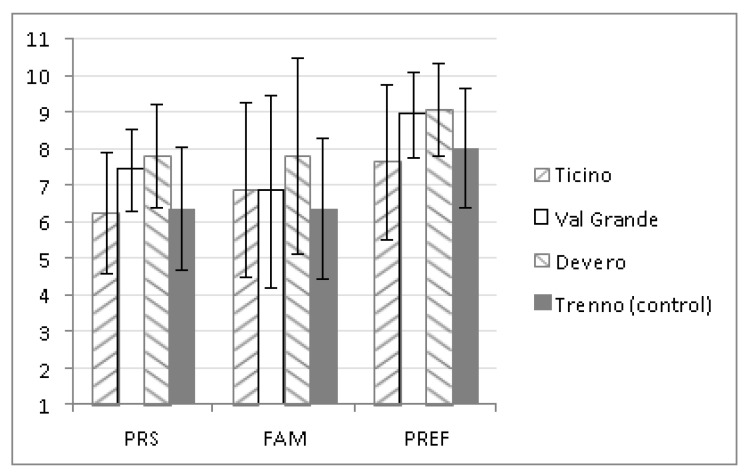
Mean scores for the perceived restorativeness scale (PRS), familiarity (FAM) and preference (PREF) (upper graph) and for the connectedness to Nature Scale (CNS) (lower graph) for each setting. A significant effect emerged for all the variables which differentiated significantly across settings following roughly this decreasing order: Devero, ValGrande, Ticino/Trenno.

**Figure 4 behavsci-08-00034-f004:**
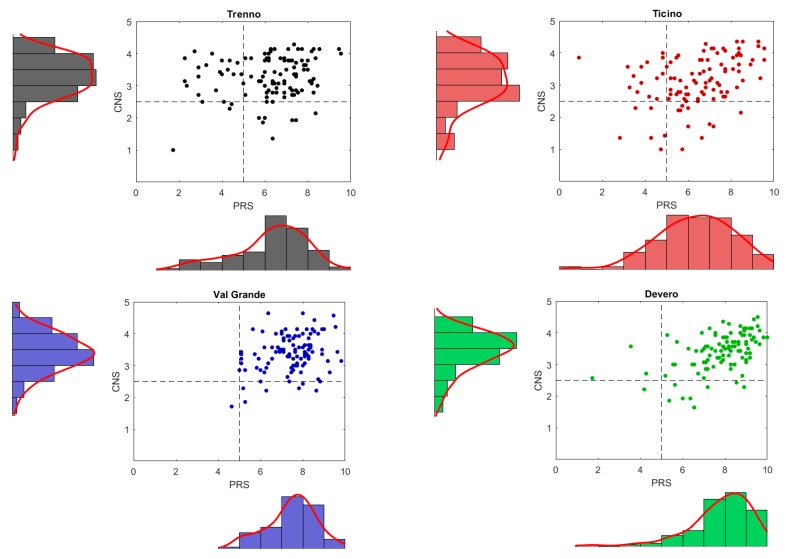
Four scatter plots, one for each setting: the x-axis represents the PRS and the y-axis represents the CNS; each axis is associated with the corresponding distribution through a histogram and a probability density function. The CNS distribution is approximately the same for all settings, while the PRS distribution lowers to the left only for low BQ settings.

**Figure 5 behavsci-08-00034-f005:**
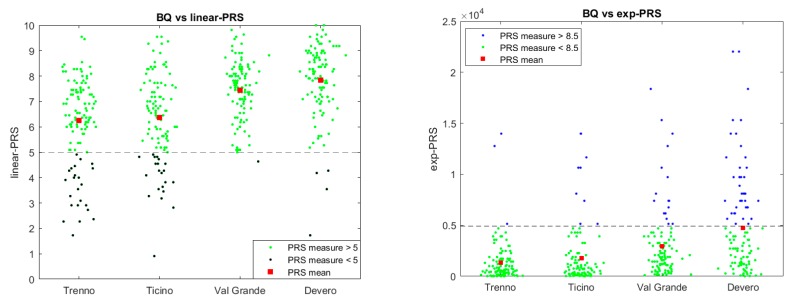
The figure on the left shows that the PRS discriminates between restorative and non-restorative settings (median cut-off score: 5). On the contrary, the scatter plot on the right points out that PRS score above 8.5, in exponential scale, is a good feature for setting classification.

**Figure 6 behavsci-08-00034-f006:**
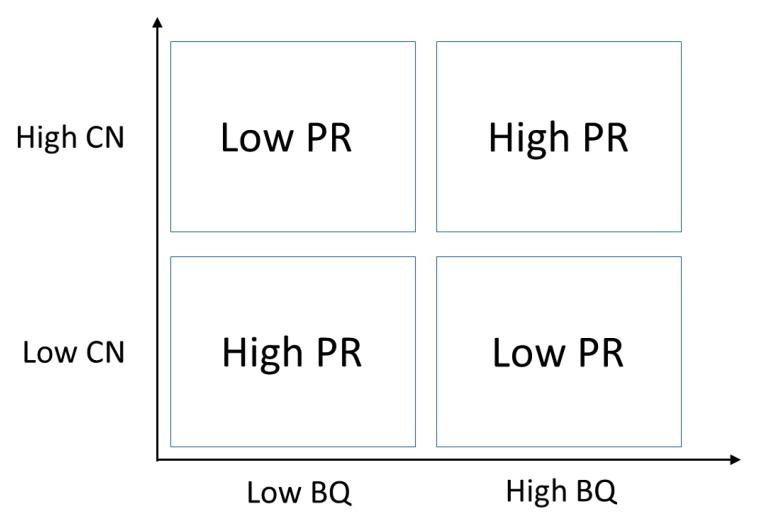
Hypothesized model where perceived restorativeness (PR) is the product of an individual’s connection to Nature (CN) and the biophilic quality of the environment (BQ). A subject’s perceived restorativeness is only high when he/she is exposed to an environment characterized by biophilic quality that exactly fits his/her own level of connection to nature.

**Table 1 behavsci-08-00034-t001:** Classification of the four settings according to the ROS classes and corresponding biophilic quality (BQ): The red line separates settings with low BQ (Trenno and Ticino, yellow background) from settings with high BQ (Val Grande and Devero, green background). Note: Ticino and Val Grande show a different level of BQ, though they offer the same level of recreational opportunity, because of their difference on the proximity model.

Proximity Model		**Recreational Potential**		
		**Low**	**Medium**	**High**
	1 Neighborhood	Trenno		
	2 Proximity		Ticino	
	3 Far		Val Grande	Devero
	4 Remote			
	5 Very Remote			

**Table 2 behavsci-08-00034-t002:** Descriptive statistics of the study sample.

	Total Subjects	Male	Female	Mean Age (SD)
Ticino	114	69	45	41.26 (16.58)
ValGrande	108	55	53	43.46 (14.73)
Devero	108	53	55	40.73 (14.43)
Trenno (control)	105	62	43	51.60 (19.66)

**Table 3 behavsci-08-00034-t003:** Mean scores (plus standard deviations) for the sensorial, symbolic, and aesthetic attributes of each setting. * = significant statistical difference among the settings, *p* < 0.05.

	Ticino	ValGrande	Devero	Trenno (Control)
Vegetation *	3.73 (0.73)	4.36 (0.69)	4.24 (0.65)	3.49 (0.73)
Diversity *	3.18 (0.75)	3.85 (0.72)	4.02 (0.79)	3.05 (0.71)
Harmony *	3.30 (0.76)	3.90 (0.72)	4.06 (0.68)	3.38 (0.68)
Openness *	3.50 (0.84)	3.64 (1)	4.26 (0.70)	4.09 (0.73)
Luminosity *	3.84 (0.77)	3.62 (0.86)	4.26 (0.87)	4.07 (0.83)
Representativeness *	3.28 (0.96)	3.85 (0.74)	4.03 (0.79)	3.05 (1.04)
Cleanliness *	2.31 (0.85)	4 (0.76)	4.03 (0.79)	3.36 (0.86)
Maintenance *	2.59 (0.76)	3.41(0.86)	4.01 (0.74)	3.50 (0.72)
Leisure activities *	3.21 (0.85)	3.19 (0.91)	3.68 (0.92)	3.92 (0.81)
Meeting place *	3.33 (0.84)	3.19 (0.84)	3.43 (0.80)	3.68 (0.93)
Novel place *	2.92 (0.84)	3.61 (0.86)	3.57 (0.80)	2.79 (0.91)
Accessible *	3.30 (0.92)	3.02 (0.94)	3.63 (0.90)	4.10 (0.79)
Safety *	2.85 (0.84)	3.41 (0.88)	3.98 (0.72)	3.32 (0.80)
Tranquility *	3.40 (0.77)	4.32 (0.72)	3.91 (0.82)	3.61 (0.74)
Crowding *	2.84 (0.92)	2.08 (0.89)	3.04 (0.98)	3.07 (0.77)
Artificiality *	1.91 (0.74)	1.69 (0.91)	1.91 (0.78)	2.61 (1.26)

**Table 4 behavsci-08-00034-t004:** Pearson’s correlations between the connectedness to Nature scale (CNS), the perceived restorativeness scale (PRS), preference (PREF), and familiarity (FAM) scores for all subjects/settings.

All Settings/All Subjects	CNS	PREF	FAM
PRS	0.34 **	0.69 **	0.34 **
CNS		0.20 **	0.12 **
PREF			0.42 **

** Correlation is significant at the 0.01 level (two-tailed).

**Table 5 behavsci-08-00034-t005:** Pearson’s correlations between the connectedness to Nature scale (CNS), the perceived restorativeness scale (PRS), preference (PREF), and familiarity (FAM) scores for each setting.

		CNS	PREF	FAM
Ticino	PRS	0.17	0.72 **	0.39 **
	CNS		0.14	0.08
	PREF			0.58 **
ValGrande	PRS	0.29 **	0.62 **	0.23 *
	CNS		0.19 *	0.21 **
	PREF			0.29 **
Devero	PRS	0.50 **	0.61 **	0.53 **
	CNS		0.23 *	0.24 **
	PREF			0.53 **
Trenno (control)	PRS	0.34 **	0.57 **	0.51 **
	CNS		0.15	0.12
	PREF			0.53 **

** = correlation is significant at the 0.01 level (two-tailed); * = correlation is significant at the 0.05 level (two-tailed).

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
