# Peer review of "An Individual’s Connection to Nature Can Affect Perceived Restorativeness of Natural Environments. Some Observations about Biophilia"

_behavsci, 2018, doi:10.3390/bs8030034_

Round 1

Reviewer 1 Report

The authors have improved the paper considerably but the methods and results are still hard to match up with the multifaceted aims of the paper. More subheadings may help with this. Specific comments are below:

Introduction

Aims of study are much clearer - good job. Also better introduction and justification of use of ROS.

Still do not understand why Nature is capitalized - this is rare in the literature, please provide justification

Methods/Results

p.p1 {margin: 0.0px 0.0px 13.3px 0.0px; line-height: 15.0px; font: 13.3px Arial; color: #000000; -webkit-text-stroke: #000000; background-color: #f6f6f6} p.p2 {margin: 0.0px 0.0px 13.3px 0.0px; line-height: 15.0px; font: 13.3px Arial; color: #000000; -webkit-text-stroke: #000000; background-color: #f6f6f6; min-height: 15.0px} span.s1 {font-kerning: none}

Please provide citations for Multi-Layer Perceptron for pattern recognition in related literature - or if not available, describe where this has been used and the fact you are now applying to new body of literature. 

L340: Unclear what “50% of the data” refers to, half the sample or half the variables?

L346: Terminology is still not consistent - here “connection to Nature” and “CN” is used. Recommend avoiding acronyms if possible.

Please provide more thorough interpretation/implications for the ANN results for readers unfamiliar with the implications of this. Also, WHY was this done? What research questions where you trying to answer? You state “precision” as outcome but it’s difficult for reader to understand (1) why more precision is needed - was there some fault/error with regressions? (2) how this technique is more precise? Is there a comparison of regression and ANN from past literature? 

The analyses should be organized around questions. Recommend descriptive subheadings in results section to guide reader what you are testing for

Figure 1 is quite low resolution and requires more dscription - figures should hold their own, in other words, readers should understand them using their captions/titles without referring to the body text.

Figure 2 no longer has picture or description for setting #1 or descriptions for #3 and #4

Figures 3 and 4 still are not obvious for readers. What is the implication of the four settings being difference from one another in Figure 3? What does “the PRS distribution lowers to the left only for low restorative settings” mean and what are the implications?

Figures 5: What are these figures, and why was a cut-off score of 5 used?

Author Response

The reviewer 1’s comments have been addressed as follow.

REVIEWER: The authors have improved the paper considerably but the methods and results are still hard to match up with the multifaceted aims of the paper. More subheadings may help with this.

OUR REPLY: We thank the reviewer for this meaningful suggestion. Now five subheadings have been added to the introduction; they are: 1.1 Perceived restorativeness from the evolutionary perspective; 1.2 Why do certain individuals seem less able to perceive the restorative properties of Nature?; 1.3 Biophilia implies affective connection to Nature; 1.4 Towards a definition of “biophilic quality” of the environment; 1.5 What is the role of familiarity on perceived restorativeness?, and the results section is now made up of  the following subheadings: 4.1 Statistical analyses; 4.1.1 Multivariate analysis of variance; 4.1.2 Pearson’s correlation; 4.1.3 Linear regression analysis; 4.1.4 Kolmogorov-Smirnov test; 4.2 Neural pattern recognition. These changes will make easier for the reader to follow the rationale behind the study, and to see the differences/similarities between the traditional statistics and the neural pattern recognition.

Specific comments are below:

Introduction

REVIEWER: Aims of study are much clearer - good job. Also better introduction and justification of use of ROS.

Still do not understand why Nature is capitalized - this is rare in the literature, please provide justification

OUR REPLY: In all previous versions of the manuscript the explanation for capitalizing the N was given in an endnote corresponding to the first time the word Nature was found in the text. By the way, here is the note:  «In this paper “Nature” is written with a capital “N” to indicate the biosphere and the abiotic matrices (soil, air, water) where it flourishes and to avoid confusion with "nature" as the intrinsic quality of a certain creature and/or phenomenon»

Methods/Results

REVIEWER: Please provide citations for Multi-Layer Perceptron for pattern recognition in related literature - or if not available, describe where this has been used and the fact you are now applying to new body of literature. 

OUR REPLY: Following the reviewer’s suggestion, a couple of citations have been added to the text. They are:

Haykin, S., Neural Networks: A Comprehensive Foundation, (3rd edition), Prentice Hall, Upper Saddle River, NJ, 2008.

Bishop, C. M. Pattern Recognition and Machine Learning, (2nd edition), Springer-Verlag New York, 2013.

The MLP is a non-linear classifier; it can be interpreted as: (1) an algorithm for general purposes that seeks the linear separation of observations through Euclidean isometries (also known as rototranslations); (2) a nonlinear weighted combination (known as a set of plastic synapses) of linear classifiers (known as perceptors or neurons) organized in a feedforward graph structure (known as a network).

Even brief, this explanation goes beyond the aim of this study, therefore we prefer to give the references without any further explanation of the method; what is still given in the text is enough.

REVIEWER: L340: Unclear what “50% of the data” refers to, half the sample or half the variables?

OUR REPLY: This request makes sense; “50% of data” refers to the 50% of results of questionnaires. Now this is made clear in the text.

REVIEWER: L346: Terminology is still not consistent - here “connection to Nature” and “CN” is used. Recommend avoiding acronyms if possible.

OUR REPLY: Ok; it was modified in the manuscript.

REVIEWER: Please provide more thorough interpretation/implications for the ANN results for readers unfamiliar with the implications of this.

OUR REPLY: More information is now available for readers unfamiliar with this method; it was made clear that MLP is a mathematical model whose objective concerns pattern recognition, which has a different granularity than the regression, therefore its results are less refined, but more reliable.

REVIEWER: Also, WHY was this done?

OUR REPLY: The ANN was performed because during the previous revision stage the statistical results were contested by you, exactly. So, we have decided to analyze the data using a totally different methodology, which confirmed the results from the traditional statistics run on the same data.

REVIEWER: What research questions where you trying to answer?

OUR REPLY: The research question is: Is there a mathematical nonlinear relationship between connection to Nature (CN) and perceived restorativeness (PR)? This important question is now part of the neural recognition pattern section.

REVIEWER: You state “precision” as outcome but it’s difficult for reader to understand (1) why more precision is needed - was there some fault/error with regressions? (2) how this technique is more precise? Is there a comparison of regression and ANN from past literature? 

OUR REPLY: Actually no “precision” was needed, the ANN was explored after your remarks about traditional statistics; accordingly, the manifold shape of the CNS/PRS relationship was analyzed more in deep. Since you questioned the reliability of statistical regressions, we decided to use a completely different technique.

However, for your interest a further explanation of the MLP follows: The MLP trained here, using a Bayesian regularization, is one of the most powerful and reliable algorithms to explore manifolds with relatively few observations (as is our study). Concerning the MLP power, Hornik et al. 19891 gave a proof of the universal approximation ability of MLPs. Regarding the Bayesian regularization and its reliability in sparse spaces, you can find most of the previous work in Neal (1996)2 . Given the reliable outcomes of the MLP, we can confirm that the previous statistical results were consistent.

[1] Hornik, K., Stinchcombe, M., White, H. (1989) Multilayer feedforward networks are universal approximators. Neural Networks 2, pp 359-366. https://doi.org/10.1016/0893-6080(89)90020-8

[1] Neal, R.M. (1996) Bayesian Learning for Neural Networks. Springer Veralg New York 1996

REVIEWER: The analyses should be organized around questions. Recommend descriptive subheadings in results section to guide reader what you are testing for

OUR REPLY: As already said above, to make easier for the reader to encompass the results now they are explained in different subheadings.

REVIEWER: Figure 1 is quite low resolution and requires more description - figures should hold their own, in other words, readers should understand them using their captions/titles without referring to the body text.

OUR REPLY: Sorry about that; now Figure 1 has been redrawn in order to accomplish a good resolution. A short description has been added as well.

REVIEWER: Figure 2 no longer has picture or description for setting #1 or descriptions for #3 and #4

OUR REPLY: Actually Figure 2 was lost when submitting the manuscript to the MDPI system.

REVIEWER: Figures 3 and 4 still are not obvious for readers. What is the implication of the four settings being difference from one another in Figure 3? What does “the PRS distribution lowers to the left only for low restorative settings” mean and what are the implications?

OUR REPLY: This comment make sense; to this end Figure 3 is now made up of two distinct graphs, one showing trends across settings for preference, familiarity and perceived restorativeness (actually all variables on 11-point Likert scales) and the other for connection to Nature (on a 5-point Likert scale); in this way the reader can get the significant differences across settings immediately.

The implication for these differences are clearly explained in the text; however variables more concerned with the setting characteristics (e.g. perceived restorativeness and preference) follow the same trend of individual’s connection to Nature, across settings.

As far as Figure 4 is concerned, you can notice that the PRS histograms have different “behaviors” (this is right term to use here). Settings with low biophilic quality -BQ (upper plots) tend to have a higher variance and a lower mean of PRS than settings with high BQ (bottom plots), the Kolmogorov-Smirnov test has confirmed this hypothesis (p <.001). Further explanations for Figure 4 are now given in the text.

REVIEWER: Figures 5: What are these figures, and why was a cut-off score of 5 used?

OUR REPLY: 5 is the median score On a 11-point Likert scale, like the PRS is; however now this is specified in the figure captioning.

Reviewer 2 Report

Overview:

The authors investigated the degree to which perceived restorativeness depends on individuals’ connection to nature and the biophilic quality of an environment. They found that people with a higher connection to nature perceive greater restorativeness. Preference and familiarity also positively predicted perceived restorativeness. These findings also depend on the biophilic quality of the environment. This research contributes to an important, and often neglected area in restorative environment research to consider the interaction between setting characteristics and individual characteristics on perceived restorativeness.

Major Comments:

I would like to see stronger evidence for the claim that “only some people consider, and are able to appreciate, exposure to natural environments as an effective and cost-free way of recovering from one’s daily hassles” and that “the average person is generally unaware of the psychological benefits that can be gained from immersing oneself in Nature” [lines 57-67]. At least one of the cited articles (Hartig, Kaiser, Bowler, and 2001) intentionally used a natural environment that they expected to be rated lower than average on perceived restorative potential. Does the stated claim hold true for environments generally rated as higher than average on perceived restorativeness? I am convinced that individual differences exist, but I am less clear on the size of these effects and the conditions under which this relationship exists.

After reading the abstract and introduction, I was left confused about how biophilic quality would vary (if at all) and what statistical model the authors would test. At first, I expected that biophilic quality of environments would vary. The authors hypothesized that perceived restorativeness depends on connectedness to nature, and that relationship depends on biophilic quality [Lines 117 – 127]. That is, biophilic quality is a moderator. In the same paragraph, it is suggested that biophilic quality will predict perceived restorativeness, and that relationship depends on connectedness to nature. That is, connectedness to nature is a moderator. At this point, I am unsure which model will be tested. Then the authors state [lines 194 – 195] that “ the present study is not interested in these variations…we want to look for differences in the level of connectedness to Nature in people visiting settings with a high biophilic quality.” This led me to think that all environments chosen for this study have high biophilic quality. But, Table 1 shows that two environments have low BQ and two environments have high BQ. One of my main concerns is I am left wondering the exact nature of how biophilic quality affects the relationship between CN and PRS. I believe this issue can be solved by being more explicit and consistent as to the nature of the relationship between biophilic quality, connectedness to nature, and perceived restorativeness, and how your statistical models appropriately test for this relationship.

Related to the above comment, there appears to be a contradiction regarding how settings differed on CN, PR, and PREF. On page 7, it is stated that “dependent variable scores differentiated significantly across settings, following roughly this decreasing order: Alpe Veglia-Devero, Val Grande, Trenno/Valle del Ticino, reflecting the biophilic quality level of the four settings”. However, on page 10, it is stated that “we notice that the trends for CN, PR, and PREF are indistinguishable between the different settings.” The latter remark matches my interpretation of Figure 3. Does CN, PR, and PREF significantly differ between all settings?

I am intrigued by the new measure of biophilic quality that the authors have developed based on the Recreation Opportunity Spectrum. However, I have questions regarding how to interpret the ROS values and BQ categories. Is the Recreation Opportunity Spectrum a linear classification system? The authors interpret environments with scores of 5-9 to have high biophilic quality, but is a score of 9 higher in biophilic quality than a score of 5? Furthermore, it is unclear to me how accessibility factors into biophilic quality. Environments with a ROS score of 7 and 9 both have high provisions, but a 7 is easily accessible (Figure 1). Should the reader interpret a BQ7 environment as having higher biophilic quality than a BQ9 environment? Table 1 implies that far proximity has higher BQ, but that conclusion is not obvious to me. I also do not understand why Trenno and Ticino are both rated as Low BQ. My intuition was that Ticino would be Moderate BQ because it is positioned centrally between the Low BQ and High BQ environments.

Figure 2 is incomplete. I assume that the peri-urban park is the missing image. It would be helpful if each picture were labeled with the park name to match the in-text description, and that landscape characteristics are provided for all four parks in the right-hand column.

Is there a purpose of collecting data on the physical and aesthetic attributes of the environments beyond obtaining descriptive information? If it is to confirm that the BQ manipulation was successful, that could be made explicit.

Was the order of questionnaires randomized or counterbalanced? It seems plausible that completing the PRS could affect Connection to Nature ratings or the physical and aesthetic attributes ratings.

I would like to see effect sizes and 95% confidence intervals reported for all analyses. In addition, please provide the post-hoc statistical support for the claim that the “dependent variable scores differentiated significantly across settings” [Line 293].

I am unfamiliar with the Multi-Layer Perceptron (MLP) technique used to test the relationship between CN, PR, and BQ. An alternative statistical technique, with which may be more familiar to readers (and I would argue is appropriate for this data) is a multilevel model. Participant ratings are nested within setting. You could also test the moderating role of BQ on the relationship between CN and PR. This analysis would allow you to test if the intercept and slope of the relationship between CN and PR varies with setting.

Minor Comments:

1. I suggest making the title more succinct by removing “Some observations about biophilia.” The remaining title captures the main result of the study.

2. [Lines 74-77] It is unclear if in this research the real-world environments were being compared to comparable virtual environments. To what does “virtual contexts” refer? Please explain what is meant by a “fuller” experience. I first interpreted this to mean more restorative but I can imagine other equally plausible interpretations. 

3. [Lines 94-96]. What is an example of an aesthetic characteristic that originated as a survival rule?

4. [Line 98] The transition to discussing “connectedness with nature” seems abrupt.  From an evolutionary perspective, would not humans be expected to have a connectedness to nature? It may be helpful to begin a new paragraph.

5. [Line 171] “thorough” should be “through”

6. [Line 192] I understand what it means for an environment to be aesthetically pleasing, but what does it mean for an environment to be “functionally pleasing”?

7. [Line 198] Remove the word “and” before “we can safely assume”.

8. I’m curious if you collected data from participants on how often they visit natural parks I general and specifically the park in which they were tested?

9. For Figure 3, are the error bars standard error or 95% confidence intervals?

10. It can be difficult to keep track of which parks are high and low BQ unless the reader is familiar with the four park names. You might consider putting the BQ value (high/low) in parentheses after the park name to make it easier on the reader.

11. [Line 313] FAM is referred to as an independent variable, which implies causality. Based on the correlational nature of the study, predictor variable would be a more appropriate term.

Author Response

We thank and appreciate the efforts made by reviewer 2 to revise our manuscript but s/he is not part of the agreement mentioned in the cover letter to the Editor-in chief.

Reviewer 3 Report

Thank you for working on this important study. It is very good to continue to explore these connections and refine our understanding of these scales, their ability to assess our experience, and the subsequent helpfulness in managing our natural resources. I have the following observations.

Line 202

The authors state that “Evidence in the literature suggests restorativeness to be a characteristic of the place, whereas connection to Nature belongs to the subject; but how can we be sure of this?”

And Line 452

 “The literature has argued that connection to Nature is a characteristic of the individual, being 452 totally independent of the environment to which they are exposed, whereas the level of perceived 453 restorativeness of a natural setting depends entirely on that environment’s characteristics.”

The very factors related to restorativeness – for example, being away and fascination are relational – that is, they emerge from characteristics of both the person and the environment. Coherence is perhaps the only attribute in the PRS 11 that could be considered a characteristic intrinsic to the place. Therefore, I suggest removing these sentences, as they do not correspond with the logic of ART of PRS. Or perhaps the authors need to rephrase what they are actually intending to convey so that I am able to understand the logic better.

Line 204

Next, the authors state, “We hypothesize that a high sense of connection to Nature helps an individual to perceive the restorative benefits of Nature, and we implicitly assume that restorativeness may also depend on the individual’s connection to Nature; at the same time, we hypothesize that a high sense of connection to Nature can be triggered by the specific characteristics of an environment, and we implicitly assume that certain environments make us feel more connected to Nature.”

This statement is more than enough and does not need to be preceded by Line 202. The hypotheses are an extension of ART and connects the ideas of biophilia and restorativeness which has been addressed well in the literature review already.

Figure 2 has pictures missing for the first setting

Line 219

Who and how many researchers classified the four settings according to the ROS measures? How was consensus reached? Which pictures were used for an analysis? How as proximity determined? Or were there ROS values listed in a previous study? If so, please provide a reference. More details are needed in the methods.

Line 228

How did you determine who was Italian? Was it Italian-speaking people because the scales were in Italian? Or are you referring to nationality, and long-term residents? More clarity is needed. By ‘end of their visits’ do you mean that the researchers were stationed at the exit and recruited participants there? Please elaborate.

Line 257

In the list of physical and aesthetic attributes, ‘meeting place’ is included, which implies social interaction. The original ROS included a factor on social encounters – was this considered when rating the settings for their biophilic quality?

Line 274

How was this counterbalance achieved?

Line 284

How was the physical and aesthetic attributes score calculated? I can see from the Table 3 that mean scores were computed – please include a sentence in the narrative also.

Line 310

Please elaborate on what you refer to as the trend in Table 3. Are there specific attributes you would refer to? If so, please highlight them. If not, provide a few sentences on your interpretation of the trend from Table 3.

Line 320 (related to Line 407)

Did you ask how many times or how often they have visited? In Line 407 you mention the importance of direct and frequent exposure to nature. Also a related thought – when does familiarity become mundane – can something become too familiar, and therefore not offer the same benefits after a while? Is this a factor in the unexpected finding reported in Line 384? What about the role of who else was along on the visit? Would my experience traveling with children who demand my attention during the visit be the same as if they had not accompanied me during the visit? That is – does my connection to nature vary according to my current frame of mind during the immersive experience? You do raise this in Lines 473 – I agree with your observation.

Line 331

I do not think including this analysis is necessary in this paper – you could perhaps report this analysis in another paper. You do not bring these results forth in your discussion – unless I am missing something important – therefore, leaving it out will be fine and will not interfere with the overall paper.

Author Response

We thank and appreciate the efforts made by reviewer 3 to revise our manuscript but s/he is not part of the agreement mentioned in the cover letter to the Editor-in chief.